# Thalamic Deep Brain Stimulation in Essential Tremor Plus Is as Effective as in Essential Tremor

**DOI:** 10.3390/brainsci10120970

**Published:** 2020-12-11

**Authors:** Julia K. Steffen, Hannah Jergas, Jan N. Petry-Schmelzer, Till A. Dembek, Tabea Thies, Stefanie T. Jost, Haidar S. Dafsari, Josef Kessler, Jochen Wirths, Gereon R. Fink, Veerle Visser-Vandewalle, Michael T. Barbe

**Affiliations:** 1Department of Neurology, Faculty of Medicine and University Hospital Cologne, University of Cologne, 50937 Cologne, Germany; Hannah.jergas@ul-koeln.de (H.J.); jan.petry-schmelzer@uk-koeln.de (J.N.P.-S.); till.dembek@uk-koeln.de (T.A.D.); tabea.thies@uk-koeln.de (T.T.); stefanie.jost@uk-koeln.de (S.T.J.); haidar.dafsari@uk-koeln.de (H.S.D.); josef.kessler@uk-koeln.de (J.K.); gereon.fink@uk-koeln.de (G.R.F.); michael.barbe@uk-koeln.de (M.T.B.); 2Department of Stereotactic and Functional Neurosurgery, Faculty of Medicine and University Hospital Cologne, University of Cologne, 50937 Cologne, Germany; jochen.wirths@uk-koeln.de (J.W.); veerle.visser-vandewalle@uk-koeln.de (V.V.-V.); 3Cognitive Neuroscience, Research Center Jülich, Institute of Neuroscience and Medicine (INM-3), 52428 Jülich, Germany

**Keywords:** essential tremor, deep brain stimulation, ET plus, tremor classification, thalamic DBS

## Abstract

The new essential tremor (ET) classification defined ET-plus (ET-p) as an ET subgroup with additional neurological signs besides action tremor. While deep brain stimulation (DBS) is effective in ET, there are no studies specifically addressing DBS effects in ET-p. 44 patients with medication-refractory ET and thalamic/subthalamic DBS implanted at our center were postoperatively classified into ET and ET-p according to preoperative documentation. Tremor suppression with DBS (stimulation ON vs. preoperative baseline and vs. stimulation OFF), measured via the Fahn–Tolosa–Marin tremor rating scale (TRS), stimulation parameters, and the location of active contacts were compared between patients classified as ET and ET-p. TRS scores at baseline were higher in ET-p. ET-p patients showed comparable tremor reduction as patients with ET, albeit higher stimulation parameters were needed in ET-p. Active electrode contacts were located more dorsally in ET-p of uncertain reason. Our data show that DBS is similarly effective in ET-p compared to ET. TRS scores were higher in ET-p preoperatively, and higher stimulation parameters were needed for tremor reduction compared to ET. The latter may be related to a more dorsal location of active electrode contacts in the ET-p group of this cohort. Prospective studies are warranted to investigate DBS in ET-p further.

## 1. Introduction

Essential tremor (ET) is characterized by bilateral upper limb action tremor but may include other neurological symptoms. In 2017, the Movement Disorder Society (MDS) published a tremor classification with new diagnostic criteria [1]. In this classification, ET with additional neurological signs, such as ataxia, rest tremor, or cognitive impairment, was defined as ET plus (ET-p). Retrospective studies suggest that ET-p is more common than ET [2,3]. However, there is an ongoing debate on the validity of the new classification [4].

Deep brain stimulation (DBS) of the thalamic ventral intermediate nucleus (VIM) and the posterior subthalamic area (PSA) is an effective treatment for medication-refractory ET. It reduces tremors up to 90% [5]. Previous studies on DBS in ET did not differentiate between ET and ET-p. Since in those studies, where patients were not systematically grouped in ET with and without plus signs, DBS response varies [6,7], this study aimed to investigate the effect of DBS in the ET-p group.

## 2. Materials and Methods

### 2.1. Patients

Patients with ET according to the previous MDS tremor classification [8], who had received VIM/PSA DBS between February 2009 and May 2019 at our center were retrospectively reclassified as either ET or ET-p. Indication for DBS and DBS implantation had been determined beforehand and independently of this study according to established criteria (medication refractory ET without contraindications for surgery, e.g., relevant cerebral atrophy, relevant cognitive impairment, …). Surgery was performed with microelectrode recordings. The final target of the DBS lead was determined after intraoperative clinical testing of effects and side effects. Retrospective classification in ET and ET-p was based on preoperative documentation (medical reports, neuropsychological testing) and, if available, on preoperative videos. Following the new MDS criteria [1], patients with plus signs were classified as ET-p. Cognitive impairment was defined as a deterioration of ≥1.5 standard-deviations in the memory domain of a neuropsychological testing battery compared to a standardized control group [9]. Inclusion criteria were DBS for at least three months before enrollment and sufficient preoperative documentation for valid classification in either ET or ET-p. In case of insufficient documentation, patients were excluded from the analysis.

### 2.2. Protocol Approval, Registration, and Consent

This investigator-initiated study was approved by the local Ethics Committee of the University of Cologne (study-no: 19-258) and conducted under the Declaration of Helsinki. Patients gave written informed consent for postoperative videos taken explicitly for this study. The study was registered at the German Clinical Trials Register (Registration -No: DRKS00021068).

### 2.3. Data Acquisition and Tremor Analysis

Age and disease duration at the time of DBS implantation were determined. Postural, kinetic, and rest tremor of upper and lower limbs, as well as head-tremor, were evaluated using the Fahn–Tolosa–Marin Tremor rating scale part A and B [10], reported as a sum score of part A and B. TRS-evaluation was based on video documentation, if available, before DBS surgery (baseline) and at follow-up in the ON-stimulation (ON, chronic stimulation with best clinical parameters) and the OFF-stimulation (OFF, stimulation in both hemispheres switched off) state. There was no defined time range between DBS implantation and follow-up videos, nor was there a standardized time from switching off stimulation and video taking in the OFF state. Most of the videos had been recorded independently of the study, implemented in our clinical routine. Additionally, patients were invited for follow-up postoperative video documentation as part of this study. TRS scores were assessed by an experienced, unblinded rater (JKS).

### 2.4. Active Contact Location and Stimulation Parameters

Locations of active contacts were determined with the LEAD DBS toolbox 2.3.2 (Berlin, Germany) [11] as described previously [12] and are reported in relation to the midcommissural point (MCP) in standardized MNI space. All contacts were mirrored to the right hemisphere. To compare stimulation settings, we calculated the total electric energy delivered per second (TEED) [13].

### 2.5. Statistical Analysis

TRS scores at baseline and follow-up (ON and OFF) were in a non-parametric distribution and were therefore compared via the Wilcoxon signed-rank test with a significance threshold of *p* < 0.05. Other pre- and postoperative scores and data on age and disease duration at the time of DBS implantation in patients with ET-p compared to ET were analyzed using the Mann–Whitney U test. The significance level of *p* < 0.05 were Bonferroni-corrected for multiple comparisons. Variance analysis was conducted using Levene’s test with a significance threshold of *p* < 0.05. Active contact location and TEED were compared via unpaired t-test or Mann–Whitney U test. Results are shown as means ± standard deviation. Statistical analyses were conducted using IBM SPSS Statistics 26 (Armonk, NY, USA) and MATLAB 2018b (Natick, MA, USA).

## 3. Results

### 3.1. Patient Characteristics

Forty-four patients (15 female) were included, of whom 31 patients (70%) were reclassified as ET-p, 13 as ET. For 17 patients (6 ET, 11 ET-p), reclassification was not only based on written documentation but also on visual evaluation of preoperative videos. The most frequent plus-sign was rest tremor (*n* = 14), followed by mild cognitive impairment (MCI, *n* = 13), dystonic posturing (*n* = 9), and gait ataxia (*n* = 5). Twelve patients showed more than one plus-sign. Patient characteristics are provided in Table 1.

### 3.2. DBS Effect and Tremor-Scores

#### 3.2.1. TRS in Stimulation-OFF and -ON

For 40 patients (28 ET-p) postoperative TRS-videos existed and TRS scores in OFF and ON were assessed from video documentation. Mean time between DBS implantation and follow-up was comparable for both groups (ET: 28.25 ± 26.20 months, ET-p: 23.68 ± 25.74 months, *p* = 0.850). TRS-OFF scores (ET: 25.83 ± 17.16, ET-p: 41.89 ± 15.56, *p* = 0.007) as well as TRS-ON scores (ET: 7.42 ± 5.20, ET-p: 15.68 ± 11.24, *p* = 0.019) were higher in ET-p than in ET, and ON scores additionally showed higher variance in ET-p (Levene test ON: F = 7.721, *p* = 0.008, OFF: F = 0.233, *p* = 0.632). Both groups showed significant (ET: *p* = 0.002; ET-p: 0.000) and comparable tremor reduction due to DBS, of 68.9% in ET (± 14.62%) and 62.0% in ET-p (± 19.60%; *p* = 0.224, Levene test: F = 1.629, *p* = 0.210, Figure 1a).

#### 3.2.2. TRS at Baseline and Follow-Up

For 17 patients (6 ET, 11 ET-p) pre- and postoperative TRS scores could be assessed. Follow-up periods were comparable (ET: 20.83 ± 17.71 months; ET-p: 13.27 ± 12.40 months; *p* = 0.256). Groups further showed comparable TRS scores at baseline (ET: 20.33 ± 5.47, ET-p: 29.27 ± 12.63, *p* = 0.180), albeit variance was again higher in ET-p (Levene test: F = 6.557, *p* = 0.022). Comparing baseline to ON DBS, significant tremor reduction was observed in both groups (ET: *p* = 0.028; ET-p: *p* = 0.006) with a comparable reduction of 75.03% (± 17.43%) in ET and 56.27% (± 22.84%) in ET-p (*p* = 0.122, Levene test: F = 0.581, *p* = 0.458). Groups showed comparable TRS scores in ON (ET: 5.00 ± 3.35, ET-p: 13.22 ± 10.22, *p* = 0.027, Levene test: F = 2.809, *p* = 0.114, Figure 1b).

#### 3.2.3. Active Contact Location

Mean coordinates of active contacts in ET were x = 11.81 ± 2.26 mm, y = −4.77 ± 1.97 mm, and z = −2.16 mm ± 1.55. In ET-p mean coordinates were x = 12.18 ± 2.51 mm, y = −4.8 ± 2.05 mm, and z = −0.63 ± 2 mm. Mean x- and y-coordinates did not differ between groups (x: *p* = 0.11, y: *p* = 0.9, Figure 1), while active contacts were located more dorsally in ET-p (*p* < 0.01, Figure 2).

#### 3.2.4. Stimulation Parameters

Stimulation parameters were available for all, but one ET-p-patient. Stimulation amplitude (ET: 1.7 ± 0.5 mA, ET-p: 2.9 ± 1.13 mA, *p* < 0.001) and TEED (ET: 30.4 ± 20.8 µJ, ET-p: 120 ± 133.2 µJ, *p* = 0.0001) were significantly higher in ET-p.

## 4. Discussion

Our data suggest that VIM/PSA DBS is effective in ET-p, with effects comparable to those observed in ET. ET-p (70.45%) was more common in our cohort, with rest tremor being the most common plus-sign. This is in line with a previous retrospective analysis reclassifying ET-patients to ET-p in 83% of cases [3]. Another study reclassified 54% of ET patients as ET-p [2]. When comparing studies on ET-p, one must consider that due to a lack of strictly defined research criteria of ET-p, inconsistencies in classification may occur, especially when comparing analyses from different study groups [4]. There is also an ongoing debate about the new tremor classification and its validity, not only because there is room for interpretation regarding certain plus signs but also because of a strong heterogeneity of ET-p-patients and a lack of consideration of the underlying etiology [4]. Further, as previous studies found ET-p-patients being older than ET-patients, the possibility remains that ET-p may represent an advanced state of ET [14]. In our cohort, VIM/PSA DBS was effective in ET-p with a mean TRS-reduction of 62.0%, comparable to the one in ET in our cohort (68.9%) and in previous ET-studies [6,15]. However, a higher TEED was applied in ET-p-patients, suggesting that a higher current was needed to achieve comparable DBS effects. This effect may be due to higher preoperative TRS scores in ET-p but could further be influenced by the location of the active electrode contacts. Several studies found that stimulation of the more ventral subthalamic area may be more efficient [16,17,18,19] so that, regarding our data, higher stimulation parameters in ET-p may partly be due to a more dorsal stimulation. The reason for the more dorsal stimulation in the ET-p subgroup is unknown and could not be investigated in this cohort in detail due to the retrospective nature of the study and the lack of electronic documentation of side effects during each clinical testing. One may speculate that more side effects (such as gait ataxia or dysarthria) occurred during ventral stimulation in the ET-p-group, potentially due to worsening of preexisting cerebellar signs. Comparing TRS scores in ET and ET-p, one must consider further that rest tremor, as a plus-sign only represented in ET-p, automatically augments TRS scores in ET-p with rest tremor and thereby in the overall ET-p subgroup. Due to the retrospective character of this study and a missing standardized time range between switching off stimulation and TRS-rating, we cannot report on a potential tremor rebound effect as it is described in previous studies [20,21]. This study further does not provide any standardized evaluation of long-term effects of DBS or tremor habituation in ET-p as only one follow-up period per patient after DBS implantation was analyzed.

## 5. Conclusions

This study shows that VIM/PSA DBS is effective in ET-p with a similar tremor suppression as observed in ET. The higher TEED needed in ET-p patients may be related to the different locations of active contacts in this cohort. For valid long-term outcomes, prospective, multicenter registries on ET-p-patients as well as defined research criteria for ET-p are needed. Future trials on DBS in tremor should include the new classification criteria for ET and ET-p.

## Figures and Tables

**Figure 1 brainsci-10-00970-f001:**
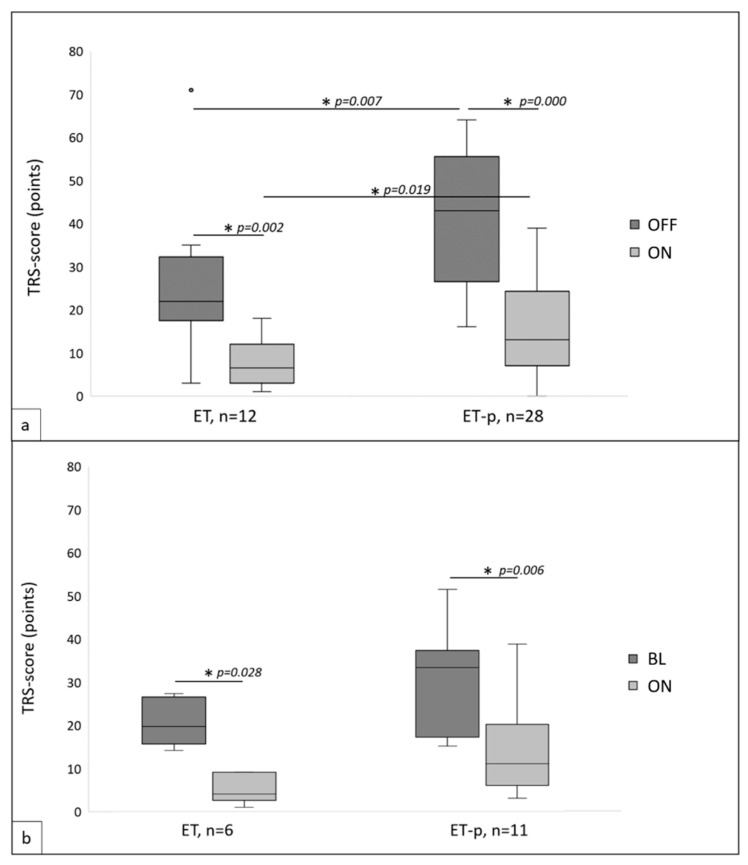
(**a**) Postoperative Fahn–Tolosa–Marin tremor rating scale (TRS) scores in essential tremor (ET) and ET-plus (ET-p) in stimulation switched OFF (OFF) and in stimulation switched ON (ON). (**b**) TRS scores in ET and ET-p at preoperative baseline (BL) and postoperative in ON. * indicates statistical significance, *p* < 0.05.

**Figure 2 brainsci-10-00970-f002:**
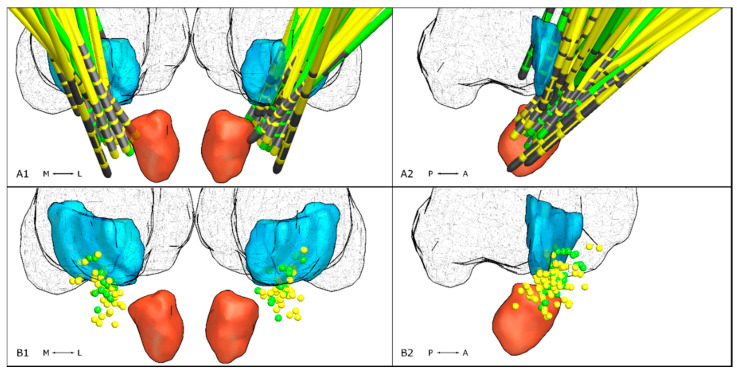
Lead locations in relation to the VIM (turquoise), the red nucleus (red) and thalamus (fishnet structure) taken from the DISTAL atlas implemented in LEAD DBS. In the top row, electrode locations of patients with ET (green) and ET-p (yellow) are shown bilaterally (**A1**) and mirrored to the right hemisphere (**A2**). Electrode contacts are marked in gray. In the bottom row, locations of active electrode contacts are shown bilaterally (**B1**) and mirrored on the right hemisphere (**B2**).

**Table 1 brainsci-10-00970-t001:** Patient characteristics and individual TRS scores. Abbreviations: f = female; m = male; age = age at DBS implantation; DD = disease duration at DBS implantation; y = years; FU = follow-up (time between DBS implantation and TRS-scoring in months, m); RT = rest tremor; MCI = mild cognitive impairment; BL = baseline; OFF = stimulation OFF; ON = stimulation ON; % (BL) = tremor reduction in percent in ON compared to baseline; % (OFF) = tremor reduction in percent in ON compared to OFF.

Pat-ID	Sex	Age (y)	DD (y); FU (m)	Syndrome	Plus-Sign	BL-TRS	OFF-TRS	ON-TRS	% (BL)	% (OFF)	Clinical Stimulation Settings Left Hemisphere	Clinical Stimulation Settings Right Hemisphere
1	f	78	9; 11	ET	-	27	-	9	66.67	-	C+, 0–100%, 60 µs, 130 Hz, 2.5 mA	C+, 8–100%, 60 µs, 130 Hz, 1 mA
2	m	67	37; 54	ET	-	16	20	5	68.75	75	C+, 2–50%, 4–50%, 60 µs, 223 Hz, 2.5 mA	C+, 10–50%, 12–50%, 60 µs, 223 Hz, 2.2 mA
3	m	71	39; 10	ET	-	22	21	3	86.36	85.71	C+, 5–34%, 6–33%, 7–33%; 60 µs; 174 Hz; 2 mA	C+, 10–24%, 11–23%, 12–23%, 13–10%, 14–10%, 15–10%; 60 µs; 130 Hz; 1.3 mA
4	m	26	9; 18	ET	-	14	3	1	92.86	66.67	C+, 3–100%, 60 µs, 174 Hz, 1.3 mA	C+, 11–100%, 60 µs, 174 Hz, 0.7 mA
5	m	56	46; 26	ET	-	17	71	9	47.06	87.32	C+, 3–100%, 60 µs, 130 Hz, 3 mA	C+, 11–100%, 60 µs, 174 Hz, 2 mA
6	m	72	11; 6	ET	-	26	19	3	88.46	84.21	C+, 1–100%, 60μs, 130 Hz, 1.1 mA	C+, 9–100%, 60μs, 130 Hz, 1 mA
7	m	64	32; 6	ET	-	-	23	7	-	69.57	C+, 0–100%, 60 µs, 130 Hz, 2.0 V	C+, 8–100%, 60 µs, 130 Hz, 1.8 V
8	m	68	30; 61	ET	-	-	33	14	-	57.58	C+, 0–50%, 1–50%, 90μs, 150 Hz, 2 V	C+, 10–100%, 90μs, 150 Hz, 2.2 V
9	m	60	44; 3	ET	-	-	30	18	-	40.0	C+, 1–100%, 60 µs, 130 Hz, 1.6 mA	C+, 9–100%, 60 µs, 130 Hz, 1.7 mA
10	m	53	44; 3	ET	-	-	17	7	-	58.82	C+, 2–34%, 3–33%, 4–33%, 60 µs, 130 Hz, 1.4 mA	C+, 10–34%,11–33%, 12–33%, 60 µs, 130 Hz, 0.7 mA
11	m	48	34; 67	ET	-	-	8	3	-	62.5	VIM 1: C+, 1–0.3 V, 60μs, 120 Hz VIM 2:2–1.1 V, 60μs, 120 Hz	C+, 9–50%, 10–50%, 60μs, 120 Hz, 1.3 V
12	m	62	29; 67	ET	-	-	35	6	-	82.86	VIM 1: C+, 0–3.1 V, 60μs, 120 Hz VIM 2: C+, 1–1.3 V, 60 μs, 120 Hz	VIM 1: C+, 9–2.1 V, 60 μs, 120 Hz VIM 2:10–2.8 V, 60 μs, 120 Hz
13	m	54	45; 18	ET	-	-	30	13	-	56.67	C+, 2–34%, 3–33%, 4–33%, 204 Hz, 50 µs, 1.3 mA	C+, 10–34%, 11–33%, 12–33%, 204 Hz, 50 µs, 1.7 mA
14	f	72	61; 12	ET-p	RT	34	-	3	91.18	-	C+, 0–100%, 60 µs, 130 Hz, 2.2 V	C+, 8–100%, 60 µs, 130 Hz, 2.4 V
15	m	22	13; 11	ET-p	Dystonia, MCI	51	-	21	58.82	-	C+, 3–30% 4–60% 5–10%, 60 µs 174 Hz, 5.7 mA	C+, 11–25% 12–75%, 60 µs 174 Hz, 4 mA
16	f	61	15; 11	ET-p	RT, MCI	37	-	20	45.95	-	C+, 2–100%, 60 us, 130 Hz, 3.9 mA	C+, 10–100%, 60 us, 130 Hz, 3.9 mA
17	m	60	29; 10	ET-p	Dystonia	21	32	10	52.38	68.75	C+, 1–30%, 2–35%, 3–35%, 60 µs, 130 Hz, 1.9 mA	C+, 13–33%, 14–33%, 15–34%, 60 µs, 130 Hz, 1.6 mA
18	m	66	16; 9	ET-p	RT	33	58	28	15.15	51.72	C+, 1–100%, 60 µs, 208 Hz, 4.2 mA	C+, 11–100%, 60 µs, 208 Hz, 2.5 mA
19	m	66	49; 43	ET-p	Dystonia	46	48	14	69.57	70.83	C+, 3–100%, 60 µs, 130 Hz, 3.8 mA	C+, 11-, 100%, 60 µs, 130 Hz, 2.9 mA
20	f	61	60; 4	ET-p	MCI	17	26	11	35.29	57.69	C+, 2–100%, 60 μs, 130 Hz, 6.5 mA	C+, 10–100%, 60 μs, 130 Hz, 4 mA
21	m	70	29; 5	ET-p	RT, MCI	16	21	11	31.25	47.62	C+; 1–100%, 60 μs, 130 Hz, 1 mA	C+, 9–100%, 60 μs, 130 Hz, 1 mA
22	f	63	21; 3	ET-p	MCI	15	20	4	73.33	80.0	C+, 1–100%, 60 µs, 130 Hz, 1.4 mA	C+, 9–100%, 60 µs, 130 Hz, 1.2 mA
23	f	77	27; 8	ET-p	RT, MCI	34	42	7	79.41	83.33	C+, 1–62%, 2–14%, 3–12%, 4–12%, 50 µs, 174 Hz, 1.8 mA	C+, 9–75%, 10–9%, 11–8%, 12–8%, 50 µs, 174 Hz, 1.7 mA
24	f	68	31; 31	ET-p	RT, Ataxia	18	25	6	66.67	76.0	2–70%, 3–30%, 5 + 100%, 60 µs, 174 Hz, 2.0 mA	10–70%, 11–30%, 13 + 100%, 60 µs, 174 Hz, 2.0 mA
25	f	70	56; 38	ET-p	RT	-	64	39	-	39.06	C+, 1–100%, 60 us, 185 Hz, 2.8 mA	C+, 9–100%, 60 us, 185 Hz, 2.8 mA
26	m	52	41; 98	ET-p	MCI	-	51	13	-	74.51	C+, 1–100%, 60 µs, 200 Hz, 2.0 V	C+, 9–100%m, 60 µs, 200 Hz, 3.3 V
27	m	77	16; 13	ET-p	RT	-	41	18	-	56.1	C+, 1–100%, 60 µs, 200 Hz, 2.0 V	C+, 9–100%m, 60 µs, 200 Hz, 3.3 V
28	f	75	10; 28	ET-p	RT	-	41	2	-	95.12	C+, 2–100%, 60 µs, 125 Hz, 3.2 mA	VIM 1: C+, 8–50% 9–50%, 60 µs, 125 Hz, 3.6 mA Vim 2: C+, 9–100%, 60 µs, 125 Hz, 3.2 mA
29	m	64	5; 15	ET-p	Ataxia	-	53	36	-	32.08	5 + 34%. 6 + 33%, 7 + 33%, 1–90%, 2–5%, 4–5% 40 µs, 204 Hz, 5 mA	13 + 34%, 14 + 33%, 15 + 33%, 9–80%, 11–10%, 40 µs, 204 Hz, 4.7 mA
30	f	65	24; 12	ET-p	Ataxia	-	33	5	-	84.85	C+, 2–21%, 3–5%, 4–29%, 5–17%,6–5%,7–23%, 60 µs, 149 Hz, 2.3 mA	C+, 10–17%, 11–26%, 12–27%, 13–10%, 14–10%, 15–10%, 60 µs, 149 Hz, 2.3 mA
31	m	58	11; 12	ET-p	MCI	-	47	22	-	53.19	C+, 0–100%, 60 µs, 130 Hz, 3.3 V	C+, 8–100%, 60 µs, 130 Hz, 3.3 V
32	m	69	6; 111	ET-p	RT	-	64	36	-	43.75	C+, 0–100%, 60 µs, 180 Hz, 3.5 V	C+, 8–100%, 60 µs, 180 Hz, 3.1 V
33	f	57	43; 32	ET-p	Dystonia	-	58	0	-	100.0	VIM 1:C+; 2–100%, 2.4 V 90μs, 120 Hz VIM 2:3–100%, 90μs, 120 Hz, 1.8 V	C+, 10–100%, 60μs, 120 Hz, 2.8 V
34	m	79	3; 37	ET-p	MCI	-	58	13	-	48.0	C+, 1–100%, 80 µs, 149 Hz, 2.7 mA	C+, 9–100%, 60 µs, 149 Hz, 2.1 mA
35	m	63	13; 17	ET-p	Dystonia, ataxia	-	54	7	-	87.04	C+ 10%, 1–100%, 2 + 30%, 3 + 30%, 4 + 30%, 50 µs, 179 Hz, 2.8 mA	C+ 20%, 12–60%, 15–40%, 10 + 19%, 11 + 19%, 13 + 21%, 14 + 21%, G + 20%, 50 µs, 179 Hz, 3.5 mA
36	m	63	44; 9	ET-p	RT, dystonia	-	52	28	-	46.15	1–100%, 2 + 34%, 3 + 33%, 4 + 33%, 60 µs, 179 Hz, 3.1 mA	9–100%, 10 + 34%, 11 + 33%, 12 + 33%, 70 µs, 179 Hz, 3.4 mA
37	m	75	66; 7	ET-p	RT, MCI	-	28	8	-	71.43	C+, 2–34%, 3–33%, 4–33%, 60 µs, 130 Hz, 2.1 mA	C+, 2–34%, 3–33%, 4–33%, 60 µs, 130 Hz, 2.1 mA
38	m	67	18; 8	ET-p	Ataxia	-	22	18	-	18.18	-	-
39	f	66	9; 14	ET-p	Dystonia, MCI	-	16	4	-	75.0	VIM 1: C+, 8–100%, 90 µs, 125 Hz, 1.3 V VIM 2: C+, 9–100%, 90 µs, 125 Hz, 2.1 V	VIM 1: C+, 1–100%, 90 µs, 125 Hz, 2.1 V; VIM 2: C+, 0–100%, 90 µs, 125 Hz, 1.3 V
40	f	77	47; 11	ET-p	Dystonia	-	16	7	-	56.25	C+, 2–100%, 60 µs, 130 Hz, 3.6 mA	C+, 10–29%, 11–43%, 12–28%, 60 µs, 130 Hz, 1.9 mA
41	f	80	20; 34	ET-p	Dystonia, ataxia	-	56	33	-	41.07	C+, 5–32%, 6–29%, 7–29% 8–10% 40 µs 170 Hz 3.3 mA	C+, 13–45%, 14–40%, 15–15% 40 µs 170 Hz 4.1 mA
42	f	70	7; 4	ET-p	RT	-	44	15	-	65.91	C+, 1–100%, 60 µs, 130 Hz, 1.2 mA	C+, 9–100%, 60 µs, 130 Hz, 1.2 mA
43	m	72	14; 25	ET-p	RT	-	64	25	-	60.94	VIM1: C+, 2–100%, 60 us, 125 Hz, 2.9 VVIM 2: C+, 1–100%, 60 us, 125 Hz, 1.8 V	C+, 8-, 100%, 60 us, 125 Hz 3.0
44	m	55	10; 26	ET-p	MCI	-	39	0	-	51.28	VIM 1: C+, 1–100%, 125 µs, 60 Hz, 0.5 V; VIM 2: C+, 2–100%, 125 µs, 60 Hz, 3.5 V	VIM1: C+, 9–100%, 125 µs, 60 Hz, 0.5 V VIM 2: C+, 10–100%, 125 µs, 60 Hz, 3.8 V

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
