# Peer review of "Thalamic Deep Brain Stimulation in Essential Tremor Plus Is as Effective as in Essential Tremor"

_brainsci, 2020, doi:10.3390/brainsci10120970_

Round 1
Reviewer 1 Report
The manuscript by Steffen et al looked to explore thalamic deep brain stimulation in essential-tremor plus patients who display neurological signs over and above the presentation of the tremor. This is an interesting manuscript of very high quality, there are only a few minor comments that need to be addressed prior to publication.
Introduction
The introduction to the manuscript is nice and clear and demonstrates the debate around the classification of essential tremor syndromes.
- Line 41, ‘it reduces tremor up to 90%’, is the superscript ‘2’ following this text a referencing error?
- Could the authors add a sentence as to why it is needed that we differentiate the DBS intervention between ET and ET-p from a clinical perspective for the reader in the second paragraph of the introduction
Materials and Methods
- A good amount of detail is provided within the materials and methods, appropriate statistical techniques were used.
Results
- The figure legend for figure 1 is not very clear, do any of these figures show post-operative scores? Or call in-stimulation post-operative. Try and keep Y axis the same in figures 1a and 1b for easier comparison
Discussion
- Do the authors have any data on the non-tremor symptoms related to ET-p post-surgery? It would be interesting to know whether the intervention worked for other symptoms
- Interesting discussion recognising limitations of diagnostic criteria
Author Response
Thank you very much for your review and you helpful comments on our manuscript.
The comments were edited as follows:
1. Line 41: There was an referencing error which was corrected.
2. Sentence why it is needed to differentiate the DBS intervention between ET and ET-p from a clinical perspective is aded.
3. Figure legend for figure 1 and layout (y-axis) is optimized.
4. Unfortunatey we do not have any data on the non-tremor symptoms related to ET-p post-surgery. We find this aspect very interesting but due to the retrospective character of the study and due to the small cohort we were not able to collect these data systematically.
Reviewer 2 Report
Thalamic Deep Brain Stimulation in Essential Tremor plus is as effective as in Essential Tremor
This is an interesting article based on the new classification of the essential tremor as two separate entities, essential tremor (ET) and essential tremor plus (ET-p). ET-p is characterized by the presence of neurological signs other than action tremor (eg, impaired tandem gait, questionable dystonic posturing, and memory impairment). The rationale of the article is that ET and ET-p are two separate entities. Some critics pointed out that the term remains uncertain. In fact, ET-plus might only denote a state condition (i.e., patients with ET might develop these additional clinical features when the disease is at a more advanced stage) (Louis et al, Lancet 2019). My principal criticism of the paper is that readers need to know if the two groups are demographically different, or we are in the presence of a more advance stage of the disease.
The other important point is the number of years of DBS treatment. A major concern regarding ventralis intermedius nucleus deep brain stimulation for essential tremor has been the loss of surgical efficacy over time in a minority of patients. Some experts have ascribed the worsening tremor to tolerance, while other evidence has suggested that disease progression (Favilla, 2012). Readers need to know not only disease duration, but also how long patients have been receiving DBS, to evaluate possible occurrence of tolerance that could account for some of the differences seen.
Otherwise, this is a very well written retrospective study with useful data that deserve to be share with the movement’s disorder community.
Some minor observations:
Comments:
Line 44 2.1. Patients:
Reviewer’s comments: Surgical criteria?
Line 47 .Indication for DBS and DBS implantation had been determined beforehand and 47 independently of this study according to established criteria:
Reviewer’s comments: Bilateral surgery dbs was done with or without Microelectrode recordings. ?
Line73 2.4. Active Contact Location and Stimulation Parameters
Reviewer’s comments: target, when is VIM when is dorsal STN?
Line95 .Patient characteristics are provided in table N1:
Reviewer’s comments: In table n°1, the reader should know the time since dbs surgery. Authors should include time from dbs. That could help the reader understand if the results are being influenced by the development of Tolerance or tremor rebound.
Line112 …with a 111 reduction of 75.03% (±17.43%) in ET and 56.27% (±22.84%) in ET-p…
Reviewer’s comments: Why was ET a bigger responder than ET-p? 1. Final position (ET more VIM and ET-p more posterior STN?) or different baseline (patients less severe?) please explain…
Line118… Mean x- and y-coordinates did not differ between groups (x: p=0.11, y: p=0.9, Fig. 1), 118 while active contacts were located more dorsally in ET-p (p<0.01, figure 2)…
Reviewer’s comments: ET-p is in the posterior subthalamic and ET is in VIM. How was decided the final target? Microelectrode recordings? How was chosen active contact during the parameters setting? This could be important for explaining the difference between final positions…
Line 134 …TableN1:
Line 158 … One might speculate that more side effects…
Reviewer’s comments: Could you include side’s effects due to DBS stimulation? Are there any tremor rebound phenomena or the development of tolerance to the stimulation?
Author Response
Many thanks for your helpfull and inspiring comments on our manuscript. The comments are answered as follows:
Line 44 2.1. Patients:
Reviewer’s comments: Surgical criteria?
This was added to the manuscript.
Line 47 .Indication for DBS and DBS implantation had been determined beforehand and 47 independently of this study according to established criteria:
Reviewer’s comments: Bilateral surgery dbs was done with or without Microelectrode recordings. ?
Surgery was done with microelectrode recordings. This was added to the manuscript.
Line73 2.4. Active Contact Location and Stimulation Parameters
Reviewer’s comments: target, when is VIM when is dorsal STN?
All patients were implanted in the VIM/PSA. None was implanted in the STN.
Line95 .Patient characteristics are provided in table N1:
Reviewer’s comments: In table n°1, the reader should know the time since dbs surgery. Authors should include time from dbs. That could help the reader understand if the results are being influenced by the development of Tolerance or tremor rebound.
Time since DBS surgery is added in table 1.
Line112 …with a 111 reduction of 75.03% (±17.43%) in ET and 56.27% (±22.84%) in ET-p…
Reviewer’s comments: Why was ET a bigger responder than ET-p? 1. Final position (ET more VIM and ET-p more posterior STN?) or different baseline (patients less severe?) please explain…
There was no statistical difference between both groups (p=0.122, line 115) so we did not discuss this slight, potential, but not statistically relevant difference.
Line118… Mean x- and y-coordinates did not differ between groups (x: p=0.11, y: p=0.9, Fig. 1), 118 while active contacts were located more dorsally in ET-p (p<0.01, figure 2)…
Reviewer’s comments: ET-p is in the posterior subthalamic and ET is in VIM. How was decided the final target? Microelectrode recordings? How was chosen active contact during the parameters setting? This could be important for explaining the difference between final positions…
Added to the manuscript (Materials and methods 2.1.)
Line 134 …TableN1:
Line 158 … One might speculate that more side effects…
Reviewer’s comments: Could you include side’s effects due to DBS stimulation? Are there any tremor rebound phenomena or the development of tolerance to the stimulation?
Added to the manuscript (discussion)